# StereoCrafter-Zero: Zero-Shot Stereo Video Generation with Noisy Restart

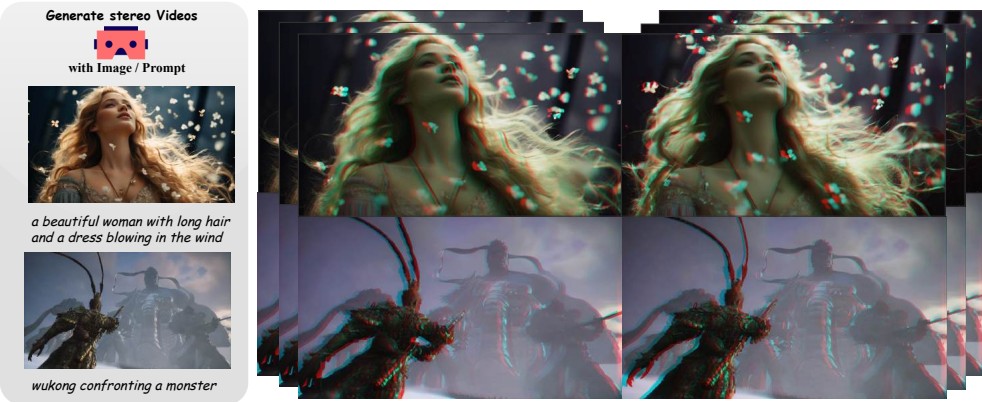

Figure 1: From only a single image and an associated text prompt as input (left), our method generates stereo video sequences visualized by an anaglyph visualization (right).

## Abstract

Generating high-quality stereo videos requires consistent depth perception and temporal coherence across frames. Despite advances in image and video synthesis using diffusion models, producing high-quality stereo videos remains a challenging task due to the difficulty of maintaining consistent temporal and spatial coherence between left and right views. We introduce *StereoCrafter-Zero*, a novel framework for zero-shot stereo video generation that leverages video diffusion priors without requiring paired training data. Our key innovations include a noisy restart strategy to initialize stereo-aware latent representations and an iterative refinement process that progressively harmonizes the latent space, addressing issues like temporal flickering and view inconsistencies. In addition, we propose the use of dissolved depth maps to streamline latent space operations by reducing high-frequency depth information. Our comprehensive evaluations, including quantitative metrics and user studies, demonstrate that *StereoCrafter-Zero* produces high-quality stereo videos with enhanced depth consistency and temporal smoothness. In terms of epipolar consistency, our method achieves an $11.7\%$ improvement in MEt3R score over the current state-of-the-art. Furthermore, user studies indicate strong perceptual gains over the previous arts, with an $8.0\%$ higher perceived frame quality and $10.9\%$ higher perceived temporal coherence. Our code will be made publicly available upon acceptance of this manuscript.

## 1 Introduction

The rapid adoption of head-mounted displays for virtual reality (VR) has created a growing demand for high-quality stereo videos, which provide immersive depth perception through paired left and right views. Given the lack of naturally acquired stereo videos, we propose exploring generative methods, particularly diffusion models, for creating these videos from scratch.

Several previous works initially focused on stereo image conversion Wang et al. (2019); Shih et al. (2020); Watson et al. (2020b); Ranftl et al. (2022), where one view is given, and its stereo pair is synthesized using techniques such as depth estimation and image warping. These approaches prioritized geometric accuracy over content generation. With the rise of diffusion models Ho et al. (2020); Rombach et al. (2022); Ramesh et al. (2022), StereoDiffusion Wang et al. (2024a) demonstrated the potential of zero-shot stereo image generation to synthesize both views simultaneously.

Recent studies have extended image stereo conversion techniques to video, emphasizing the importance of temporal consistency to produce smooth and coherent stereo video sequences Shi et al. (2024a); Zhao et al. (2024). Yet, zero-shot stereo video generation remains unexplored. This task is inherently more complex than image generation, as it requires meticulous handling of depth cues to produce realistic parallax and consistent depth perception in both views Barron & Popović (2015), while maintaining temporal coherence across frames and inter-view consistency between stereo pairs. Advanced video depth estimation models have improved temporal consistency by accurately capturing and maintaining depth information across video frames Luo et al. (2020); Kopf et al. (2021); Hu et al. (2024); Chen et al. (2025). They played a crucial role in enhancing the overall visual fidelity and temporal smoothness of the output of existing video generation models. However, these methods lack mechanisms for dynamically adapting depth information during the diffusion process, which is necessary to accurately represent scene dynamics in stereo video generation.

To address these challenges, we introduce **StereoCrafter-Zero**, a novel framework for zero-shot stereo video generation that leverages video diffusion priors to generate high-quality stereo videos without the need for paired training data. The straightforward solution to this problem is to break it down into two known components: video generation and stereo video conversion. Instead of pixel-level generation and conversion, we propose a stronger coupling of generation and conversion by improving and refining the consistency within the latent features of a video diffusion model. One significant advantage is that our method does not require precise disparity maps. We introduce the concept of *dissolved depth maps*, which retain only the low-frequency structural depth information. Our key insight is that latent-space warping benefits more from coarse geometry than fine-grained depth. Unlike image-space warping, where high-frequency depth helps preserve pixel-level accuracy, latent-space warping prioritizes coarse geometry and semantic consistency without requiring accurate depth maps. Furthermore, our approach seamlessly integrates advanced video depth estimation into the diffusion-based synthesis process, ensuring both depth consistency and temporal coherence. Our main contributions are as follows:

- We enhance latent consistency in stereo generation by employing *noisy restart* to create stereo-aware initial latents, followed by *iterative refinement* that systematically injects controlled noise into the diffusion process, progressively improving the harmony of the latent space.
- We introduce *dissolved depth maps*, a novel depth representation that retains only low-frequency structural depth while suppressing high-frequency information, effectively reducing artifacts in the generated right view.
- We perform thorough evaluations, including statistical analysis and user studies, to validate the ability of our method to generate high-quality stereo videos. Our approach achieves a new state-of-the-art in epipolar consistency, with user studies confirming significant improvements in visual clarity and temporal smoothness.

## 2 RELATED WORKS

### 2.1 NOVEL-VIEW SYNTHESIS

Novel-view synthesis task aims to generate images from new camera perspectives based on one or more source images. Recent novel-view synthesis works such as Li et al. (2023); Liu et al. (2023); Yu et al. (2023); Tang et al. (2023); Sun et al. (2023); Yu et al. (2024b); Bai et al. (2024) demonstrate good results in creating a stereo pair of a given scene. However, these methods require scene-specific optimization, which limits their applicability to video data. Another category of approaches Shriram et al. (2024); Chung et al. (2023); Zhang et al. (2024b) employs the depth-warping technique to synthesize novel views and subsequently refines the warped images. These approaches suffer from visual artifacts in inpainted regions, particularly in complex scenes with large disparities or occlusions. More importantly, these methods cannot enforce temporal consistency, which is an important requirement for handling video-based novel-view synthesis. The Collaborative Video Diffusion (CVD) technique (Kuang et al., 2024) utilized a cross-video synchronization module to

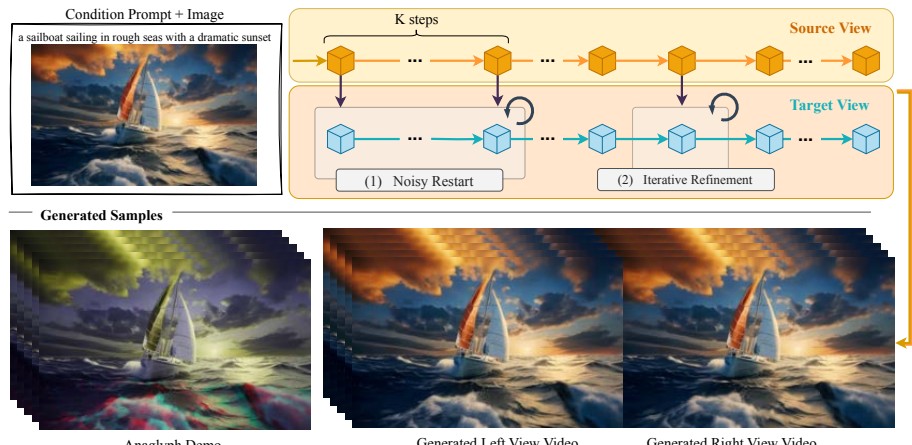

Figure 2: An overview of the *StereoCrafter-Zero* pipeline. Top: Our method is based on two main components: (1) **Noisy Restart** for a robust initial latent estimation (Sec. 3.1) and (2) **Iterative Refinement** for the latent refinement (Sec. 3.2). These components are applied to the target view latents (blue tensors) to achieve temporal coherence and inter-view consistency with the source view latents (orange tensors). Bottom: The proposed pipeline takes an image and text prompt as input, generating left and right views that produce a strong stereoscopic effect.

directly generate multi-view videos for predefined camera trajectories. Another line of work (Yu et al., 2024a; Bahmani et al., 2024; Zhang et al., 2024a) focuses on explicit 3D scene construction with temporal dynamics, enabling the generation of novel-view videos. However, these methods are limited to simple object-centric scenes and exhibit limited visual quality when applied to complex scenes. Generating novel-view videos with complex visual content remains an open challenge.

## 2.2 STEREO CONTENT GENERATION

Stereo content generation has evolved significantly from traditional disparity estimation and view synthesis methods to modern deep learning approaches that enhance both depth accuracy and visual realism. Early approaches Xie et al. (2016); Wang et al. (2019) utilized neural networks and generative networks to predict disparity maps and synthesize the corresponding stereo image pairs. Recent advances have extended these techniques to video by incorporating temporal coherence, thereby creating smooth and immersive stereo video sequences Zhang & Wang (2022); Shi et al. (2024a); Zhao et al. (2024). Notably, diffusion models have been successfully adapted for zero-shot stereo image synthesis Wang et al. (2024a). However, applying these models to stereo video generation remains challenging, primarily due to the need to maintain both spatial depth fidelity and temporal consistency across frames. A concurrent work Dai et al. (2024) proposes a *frame matrix* approach, which involves placing multiple camera views and warping the latent space on every DDIM (Denoising Diffusion Implicit Models) sampling step. In contrast, our method generates new viewpoints without the need for warping at every step, which significantly reduces the computational costs.

## 3 METHOD

This section describes *StereoCrafter-Zero*, which leverages video diffusion priors for zero-shot stereo-consistent video generation. Let $\mathbf{X} = \{x_T, x_{T-1}, \ldots, x_0\}$ denote the DDIM latent sequence from a video diffusion model with $T$ diffusion steps (e.g., $T = 50$). Each latent $x_t \in \mathbb{R}^{B \times C \times T \times H \times W}$ is a five-dimensional tensor representing a batch of video frames, where $B$ is the batch size, $C$ the number of channels, $T$ the temporal dimension (number of frames), and $H$ and $W$ the spatial height and width, respectively. By decoding $x_0$, we obtain a video sequence $V \in \mathbb{R}^{B \times 3 \times T \times H' \times W'}$, where $H'$ and $W'$ are the corresponding height and width of the decoded videos. To capture the scene geometry and enable accurate latent warping, we generate depth maps $\mathbf{D} \in \mathbb{R}^{B \times T \times H' \times W'}$, which provide per-pixel, temporally consistent depth information. Using these depth maps, the latent features are warped to achieve stereo consistency and account for parallax

effects. Let $\Delta$ denote the disparity map, which represents the horizontal shift between the left and right stereo views. The warp operation $\mathbf{W}(x, \Delta)$ (see Eq. (7) in the supplementary) is then applied to produce the warped latent representation $x_t^{\text{warp}} = \mathbf{W}(x_t, \Delta)$. Details of our efficient implementation (**1000 $\times$ faster** than a traditional non-vectorized warping method) of the warping algorithm are provided in the supplementary. This warping aligns the latent representations according to depth-induced disparities, which is essential for generating realistic stereo views. The warping process, however, introduces blank regions $x^{blank} \in \mathbb{R}^{B \times C \times T \times H \times W}$ in the warped latents due to occlusions or disocclusions, defined as:

$$x_t^{blank} = 1 \text{ if } x_t^{\text{warp}} \text{ is undefined,} \qquad 0 \text{ otherwise.} \qquad (1)$$

With the aid of a disparity map, each latent feature can be decomposed into $x_t^{blank}$ and $x_t^{\text{warp}}$ after warping. Maintaining consistency between these two latent parts is critical for creating meaningful and harmonized outputs for each individual view with correct depth cues. In the diffusion process, $\epsilon_t$ is introduced as a random noise term serving as the sole source of randomness (see Eq. (8) in the supplementary). By iteratively injecting $\epsilon_t$ into the diffusion steps, we refine the latent features, promoting coherence and fidelity in the generated results.

Our paper aims to improve the consistency between $x_t^{blank}$ and $x_t^{\text{warp}}$. We achieve this through two key steps: (1) **Noisy Restart** (Sec. 3.1), which initializes a reasonable $x^{blank}$, and (2) **Iterative Refinement** (Sec. 3.2) that improves the consistency between $x^{blank}$ and $x^{warp}$. In addition, we introduce **Dissolved Depth Maps** (Sec. 3.3), which transform depth maps into lower-frequency representations to enhance the consistency of warped latents with the video diffusion prior.

## 3.1 NOISY RESTART

The noisy restart mechanism operates over $K$ diffusion steps for $L$ iterations. We denote the $K$ diffusion steps as $\{x_t\}_{t=0}^K$, where $x_t$ represents the latent state at step $t$. While $K$ can be a set of discrete timesteps, we use $K = \{49, ..., 45\}$ in our implementation. The main purpose of the noisy restart is to introduce random noise into the latent space to prevent structural repetition and ensure smooth transitions. In the first iteration, $K$ filling latents are generated for each diffusion step, while the subsequent iterations will refine the obtained $K$ latents. Referring to Eq. (8) in the supplementary, a random noise $\epsilon_t$ is introduced at each sampling step. If each iteration uses different noise values, consistency across iterations can be disrupted. To address this issue, we initialize fixed noise tensors $\epsilon'_t$ before starting the diffusion sampling sequence and use the same random seed for each iteration. This operation ensures that the random generator status remains the same across multiple iterations, and the noise injection is controlled solely by $\epsilon'_t$ and $\alpha_t$ in Eq. (2). Here $\alpha_t$ is a weighting factor that regulates the relative contributions of the previous latent state and the injected noise. Meanwhile, such a controlled approach promotes structural stability throughout the generation process, ensuring consis-

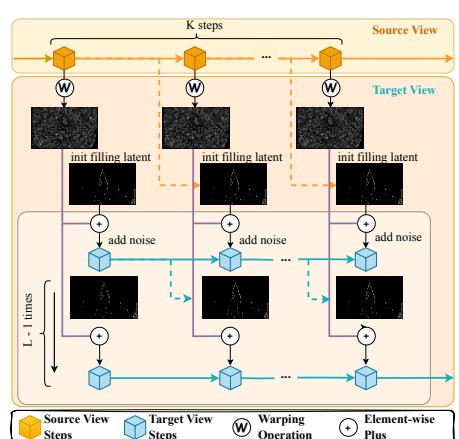

Figure 3: Illustration of the noisy start strategy. At selected steps, we replace the target view sampling with a warped source view. Occluded/disoccluded areas are then filled using the non-warped source view latent, with added noise injected into the latent space. Subsequent iterations update the latents with values from the preceding iteration, while preserving the non-occluded regions.

tent low-frequency noise that preserves major features across frames. In our work, we used $L = 7$ iterations, with the first iteration directly using the left-view latents as the filling latents. Noise is injected into the latent state through a weighted addition, balancing the contributions of the existing latent and the injected noise. The update equation for $x_{t-1}$ is given by:

$$x_{t-1} = x_t \cdot (1 - \alpha_t) + \sigma_t \epsilon'_t \alpha_t \qquad (2)$$

where $\sigma_t$ is the noise magnitude at timestep $t$, scaling the impact of injected noise, while $\alpha_t$ is the balancing coefficient to control the mixture between the latent state and the noise. This approach significantly enhances stereoscopic effects, as shown in Fig. 4.

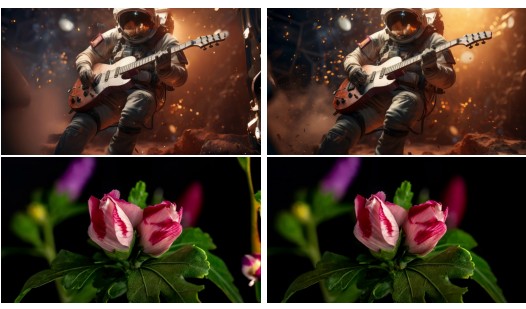

(a) w/o Noisy Restart ($L = 1$)     (b) w/ Noisy Restart ($L = 3$)     (c) w/ Noisy Restart ($L = 7$)

Figure 4: Impact of the Noisy Restart on stereo effects. This anaglyph visualization vividly demonstrates the improvement. Increasing noisy restart iterations strengthens the stereo effects. Note: Results are achieved solely with noisy restart.

(a) w/o border handling     (b) w/ border handling

Figure 5: Abrupt border handling. (a) Images with noticeable abrupt artifacts along the right edge. (b) Border artifacts are effectively removed.

**Abrupt Border Handling.** Direct warping of the right view can result in blank regions along the right border. These blank areas may create irrelevant or distracting content in the border regions during the iterative process. To address this issue, we introduce a border-cleaning function that selectively masks and fills these regions using information from the left view, ensuring visual coherence. A border mask, $M_{\text{border}}$, is created using a heuristic function to select the border areas. We then inpaint these areas with the corresponding regions in the left-view latents, denoted as $L_l$. The resulting border refined latent $L_{r,\text{refined}}$ is defined as:

$$L_{r,\text{refined}} = L_r \left(1 - M_{\text{border}}\right) + L_l \, M_{\text{border}}. \tag{3}$$

This operation replaces the blank columns in the right view with the content from the left view, ensuring visual continuity and eliminating the reasonable but distracting artifacts of the stereo pair. A qualitative evaluation of this technique is presented in Fig. 5.

## 3.2 ITERATIVE REFINEMENT

To optimize computational resources, we limit noisy restart to initial sampling steps, where it has the highest impact on stereo effects. For later stages, we introduce Iterative Refinement, which leverages video diffusion priors to enhance details in occluded regions. It operates at specific diffusion steps, repeating the denoising operation $N$ times on the occluded regions. For each refinement step, we first obtain a predicted latent $x_t^{j=1}$ with the UNet denoiser $\epsilon_\theta$.

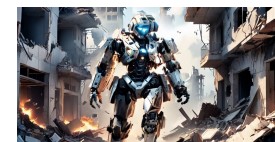 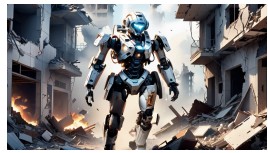

(a) w/o Iterative Refine     (b) w/ Iterative Refine

Figure 6: Impact of Iterative Refinement. Without it, warping artifacts may degrade the filling areas.

For subsequent iterations ($j > 1$), the latent is updated as follows:

$$x_t^j = (1 - M)\, x_t^{(j-1)} + M \; \epsilon_\theta\!\left(x_t^{(j-1)}\right), \qquad M \text{ is the mask for occluded regions.} \tag{4}$$

This approach optimizes computational resource usage for efficient refinement while delivering substantial quality improvements, as can be seen in Fig. 6.

## 3.3 DISSOLVED (LOW-FREQUENCY) DEPTH MAPS

Depth estimation models are typically designed to capture fine, detailed depth maps. However, unlike image-space warping, where high-frequency depth helps preserve pixel-level accuracy, latent-space warping prioritizes coarse geometry and semantic consistency. When warping the latent space of a diffusion model, these high-frequency details can compromise the latent space consistency. As shown in Sec. 4.3, high-precision depth maps often introduce artifacts, such as ghosting, during the

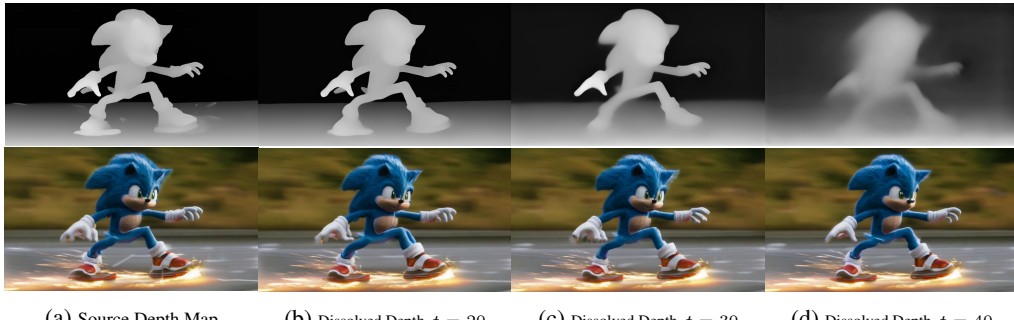

(a) Source Depth Map     (b) Dissolved Depth, $t = 20$     (c) Dissolved Depth, $t = 30$     (d) Dissolved Depth, $t = 40$

Figure 7: Dissolved depth maps from *DepthCrafter* (50-step schedule). The top row shows the gradual removal of high-frequency details. The bottom row shows a reduction of ghosting effects.

warping process. We suspect that this fine-grained warping, especially the sharp edges within depth maps, interfere with temporal coherence, making them less compatible with video diffusion priors.

To address this issue, we propose a depth-dissolving technique that transforms the depth maps into a lower-frequency representation. Inspired by the semantic simplification technique proposed by *Dissolving Is Amplifying (DIA)* Shi et al. (2024b) and Wang *et al.* Wang et al. (2024b), we generate **dissolved depth maps** by leveraging the inherent properties of diffusion models to act as a low-pass filter on the latent space. Using a diffusion-based depth estimation model, we first obtain a depth latent $x_T$ at the final diffusion step $T$. Instead of performing a full reverse diffusion process, we only execute a single-step reverse diffusion on $x_T$. This approximation is designed to suppress high-frequency details, thereby reducing noise and artifacts in the latent representation. Formally, we denote the approximated initial state as $\hat{x}_{t\to0}$, which depends on the selected time step $t$. The process is defined as follows:

$$\hat{x}_{t\to0} = \sqrt{\frac{1}{\bar{\alpha}_t}} \cdot x - \sqrt{\frac{1}{\bar{\alpha}_t} - 1} \cdot \epsilon_\theta(x, t), \tag{5}$$

where $\bar{\alpha}_t$ represents the cumulative product of the diffusion coefficients up to time $t$, $\epsilon_\theta$ is the predicted noise at time $t$. By emphasizing global structure over pixel-level depth variations, our approach enables smoother disparity transitions. As a result, the warped latents maintain better temporal coherence and align more effec-

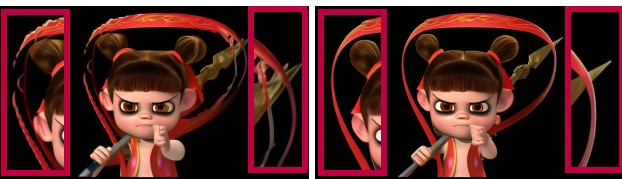

(a) w/o Dissolved Depth     (b) w/ Dissolved Depth

Figure 9: Dissolved depth effectively reduce the artifacts.

tively with the video diffusion priors. Experimental results confirm our hypothesis that dissolved depth maps reduce artifacts such as ghosting and staircase effects. We illustrate a representative case in Fig. 9, where dissolved depth maps can significantly reduce artifacts such as ghosting and jaggies. We provide additional experiments and visual illustrations of the impact of dissolved depth maps on the stereo effects in the generated stereo videos in our supplementary materials.

## 4 EXPERIMENTS

### 4.1 IMPLEMENTATION DETAILS

We implement our method based on *DynamiCrafter* Xing et al. (2024), a state-of-the-art diffusion-based video generation framework. We also evaluated our method with other diffusion-based video generation methods in our supplementary material. We infer video depth maps using *DepthCrafter* Hu et al. (2024). We apply the cross-view attention mechanism on all sampling steps apart from the cross-attention between the latents and the conditions. We use $L = 7$ for noisy restart from $t = 49$ to $t = 45$. Afterward, warping is performed to update the side-view latents every 5 steps, followed by iterative refinement at each warping stage. We found that the last iterative refinement step of $t = 15$ is sufficient for most cases. The number of refinements is set to $N = 4$.

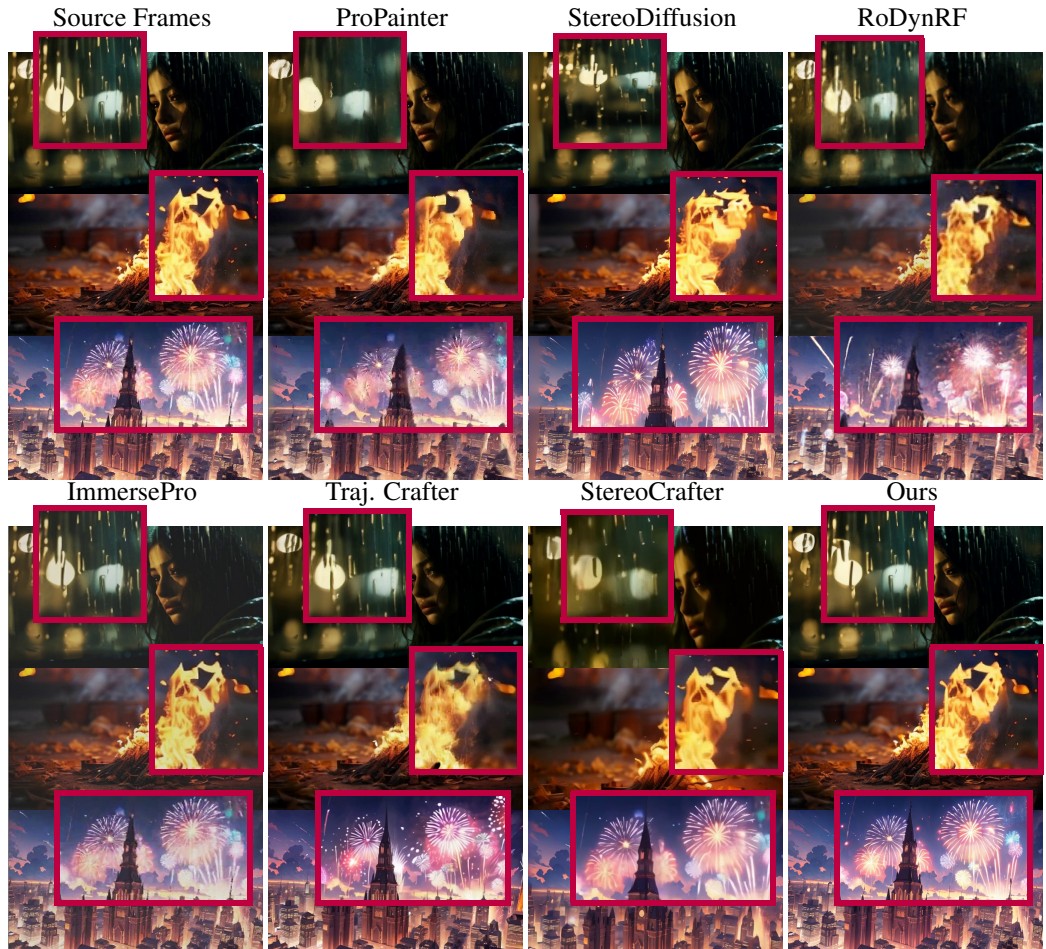

Figure 10: Visual comparison with *ProPainter*, *RoDynRF*, *ImmersePro*, *TrajactoryCrafter*, *StereoCrafter*, and *StereoDiffusion*. Low resolution methods such as *ImmersePro* and *StereoCrafter* produce videos with less fine details. *ProPainter* can hardly handle the fine details such as rain drops and fire flames, *RoDynRF* can hardly maintain the scene structure correctly, *StereoDiffusion* may produce more distortion since there are no temporal constraints (stronger when view in videos), and *TrajctoryCrafter* can hardly handle complex scenes. We provide video samples in supplementary.

## 4.2 RESULTS

**Baseline Methods.** Since there are no direct competitors for zero-shot stereo video generation, we compare our method against methods including *ProPainter*, *RoDynRF*, *ImmersePro*, *StereoDiffusion*, *TrajectoryCrafter*, and *StereoCrafter*. For benchmarking purposes, we generated 40 video clips with *DynamiCrafter* spanning diverse domains such as anime, human subjects, animals, various objects, and imaginary contents. Note that generating one video clip with *RoDynRF* requires approximately 10 hours on an NVIDIA A100, we therefore neglected it in our benchmark tables but presented a visualization in Fig. 10.

**Quantitative Results.** Unlike stereo conversion, the stereo generation task lacks a ground-truth right view for direct comparison. Therefore, our evaluation focuses on two key aspects: semantic consistency and multi-view consistency between the left and right views of the generated video. For semantic consistency, we follow prior works in novel-view rendering Cai et al. (2023; 2024); Kuang et al. (2024), and report CLIP-F metric Zhengwentai (2023) to compute the consistency between the source-view videos and their corresponding generated views. For multi-view consistency, we employ MEt3R Asim et al. (2025) to assess the epipolar consistency between the left and right views. We use *DINOv2* Oquab et al. (2023) as the feature extractor for MEt3R.

Tab. 1 shows the performances against the baselines. In general, low-resolution video conversion methods such as *ImmersePro* and *StereoCrafter* produce better *MEt3R* scores. We suspect this is

due to the lower resolution depth maps, which come with fewer distracting fine depth details. Our approach leverages this insight directly. By using dissolved depth maps, we selectively remove these potentially distracting details from the depth information without downsampling the video itself. Thus, our method is capable of producing high-resolution videos without compromising the epipolar consistency. Further evaluations such as varying baseline settings and different depth models are provided in our supplementary.

Table 1: Benchmark results. The best and second best results are highlighted in red and cyan colors, respectively.

|  | ProPainter | ImmersePro | StereoDiffusion | Traj. Crafter* | StereoCrafter | Ours |
|---|---|---|---|---|---|---|
| CLIP-F ↑ | 96.45 | 96.99 | 91.09 | **97.43** | 93.59 | 97.16 |
| MEt3R ↓ | 8.82 | 5.69 | 6.09 | 6.21 | 5.61 | **4.95** |

*: *TrajectoryCrafter* uses the first 10 frames for camera pose estimation. We, therefore, tripled the frames. The metrics are computed with frames downsampled back.

**User Study.** To evaluate the quality of the generated videos, we conducted a single-blind user study involving 29 participants using a Meta Quest 3 headset. Participants were unaware of the method used to create each of the videos. Each participant evaluated the quality of 5 selected videos generated by our method and other baseline methods. The results of this user study are summarized in Tab. 2, where our method achieved the highest overall user rating. Although *TrajectoryCrafter* was not originally designed for stereo generation, it nevertheless delivered the second-best performance, particularly demonstrating strong stereoscopic effects.

Table 2: User study results comparing preference scores for different criteria. Scores are on a scale from 1 to 5, with the highest and second-highest values highlighted in red and cyan, respectively.

|  | ProPainter | ImmersePro | StereoDiffusion | Traj. Crafter | StereoCrafter | Ours |
|---|---|---|---|---|---|---|
| Frame Quality | 3.27 | 3.20 | 3.12 | 3.75 | 3.38 | **4.05** |
| Temporal Coherence | 3.34 | 3.42 | 2.83 | 3.57 | 3.38 | **3.96** |
| Stereoscopic Effects | 3.27 | 3.50 | 2.75 | **3.79** | 3.52 | **3.79** |
| Overall Conformity | 3.20 | 3.34 | 2.83 | 3.83 | 3.46 | **3.98** |

**Qualitative Results.** Fig. 10 presents visual comparisons against various competing methods, and stereo video examples are included in the supplementary material. The results show that our method consistently generates high-quality stereo videos, outperforming other approaches in terms of temporal coherence, resolution, and stereo effects. In general, *ProPainter* struggles to accurately reconstruct fine details (*e.g.* shifted raindrops). *RoDynRF* fails to maintain the structure of the scene during view changes. *StereoDiffusion* introduces distortions due to the lack of temporal consistency, while *ImmersePro* can alter the scene brightness with weaker stereo effects. Recent *TrajectoryCrafter* may wrongly interpret complex scenes, and *StereoCrafter* uses downsampled videos, which removes fine details. We strongly encourage readers to watch the videos in the supplementary material, since temporal coherence and spatial jittering are hard to fully appreciate in static images.

## 4.3 DISCUSSION

**Why Depthcrafter?** As the essential ingredient for creating stereoscopic effects, depth information significantly affects the quality of the generated right views. Besides DepthCrafter, we evaluated our method using several state-of-the-art depth models, including Depth Pro Bochkovskii et al. (2024), Depth Anything Yang et al. (2024), and Video Depth Anything Chen et al. (2025), to assess the impact of depth estimation accuracy on stereo generation. For image-based depth estimation models, we applied a disparity propagation algorithm (see supplementary) to enhance the temporal consistency. Our method is compatible with any depth method. However, the depth maps are processed by a depth dissolving technique. This technique is implemented as the reverse diffusion process in the diffusion latent space. This reverse diffusion process comes from the pre-trained Depthcrafter architecture. We conjecture that this makes the distribution of depth latents

Table 3: Performances with different depth models. Without the depth dissolving technique, similar performances are observed for different depth estimation models.

|  | D. Pro | D. Anything | V. D. Anything | D. Crafter w/o dsl. | D. Crafter w/ dsl. |
|---|---|---|---|---|---|
| MEt3R ↓ | 6.78 | 6.71 | 6.79 | 6.70 | **4.95** |

from DepthCrafter more compatible. As shown in Tab. 3, without the depth dissolving technique, similar performances can be observed for different depth estimation models. However, depth dissolving gives a clear advantage to Depthcrafter ( $26\%$ improvement on MEt3R from 6.70 to 4.95). Additional results exploring the impact of varying dissolving levels are provided in the supplementary material.

**Noisy Restart and Iterative Refinement.** Noisy restart selectively injects controlled noise into disoccluded regions during early diffusion steps, shaping global stereo disparity and structural coherence. Iterative refinement performs targeted re-denoising ($L = 11$) at specific steps without noise reintroduction, harmonizing filled regions with warped latents. By varying the restart window $K$ and the number of denoising rounds $L$, we found that increasing $K$ in later sampling steps degrades performance, as shown in Tab. 4. In Tab. 5, by using the optimal settings ($K = 6, L = 7$), we vary the number of refinement rounds $N$. This experiment shows that moderate refinement ($N = 4$) achieves the best performance.

Table 4: Ablation on restart parameters $K$ (window) and $L$ (rounds).

| $K$ | $L$ | Total Steps | MEt3R ↓ |
|---|---|---|---|
| 6 | 5 | 30 | 0.0525 |
| 6 | 7 | 42 | **0.0513** |
| 6 | 9 | 54 | 0.0546 |
| 11 | 5 | 55 | 0.0555 |
| 11 | 7 | 77 | 0.0545 |
| 11 | 9 | 99 | 0.0580 |
| 21 | 5 | 105 | 0.0609 |
| 21 | 7 | 147 | 0.0607 |
| 21 | 9 | 189 | 0.0620 |

Table 5: Ablation on iterative refinement rounds $N$ using $K = 6, L = 7$.

| $K \times L$ | $N$ (Rounds) | Total Steps | MEt3R ↓ |
|---|---|---|---|
| 6×7 | 0 | 42 | 0.0513 |
| 6×7 | 2 | 56 | 0.0524 |
| 6×7 | 4 | 70 | **0.0495** |
| 6×7 | 6 | 84 | 0.0523 |
| 6×7 | 8 | 98 | 0.0535 |

The reported "total steps" in both tables correspond to the cumulative number of diffusion steps, which can be seen as a proxy for the runtime. Our method runs in total for 150 diffusion steps for stereo video generations, corresponding to threefold total computation compared to a common monocular video generator (*e.g.* 50 steps).

**Noise-Injection For Latent Refinement.** Our noise injection strategies are conceptually related to the noise re-injection mechanism introduced in Time Reversal Fusion (TRF) Feng et al. (2024). TRF adopts a global noise perturbation strategy, whereas we use a stereo-aware, region-selective noise injection. Our experiments indicate that stronger noise injection during noisy restart consistently produces a clearer and more stable stereo effect, as shown in Fig. 4. Yet both approaches reach a similar conclusion, that small perturbations have minimal effect at early denoising stages. This behavior is well grounded in diffusion dynamics Ho et al. (2020), where the variance term $\beta_t$ shrinks toward later timesteps, reverse-process updates become too small to correct earlier structural choices. As a result, we apply stronger, stereo-targeted noisy restart in the early timesteps, ensuring that the model converges toward a stable and geometry-consistent solution.

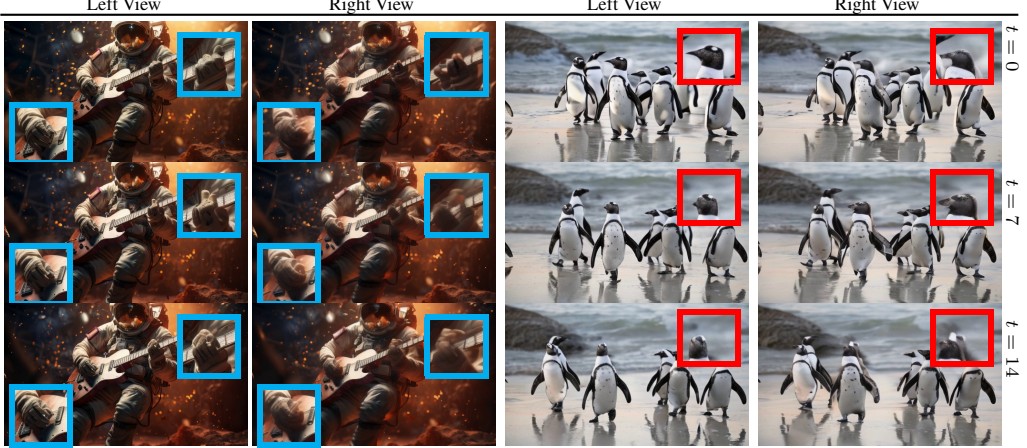

Figure 11: Demonstration of the failure cases. Our method can fail in the strong motion areas, such as the rapidly moving hand gesture and head pose.

As observed in Repaint Lugmayr et al. (2022), image inpainting models require sufficient diffusion steps to fully harmonize conditional inputs before the noise variance $\beta_t$ becomes too small in later timesteps. Otherwise, the model loses the capacity to correct structural inconsistencies. We observe an analogous phenomenon in our video generation method. Warping depth maps at earlier diffusion stages only is better than warping depth maps at all diffusion stages. It is intuitive that a warping operation alters the distribution of RGB video latents to a distribution that is slightly out of the domain for the diffusion model. Therefore, there is a trade-off between warping high-frequency details at later diffusion stages and moving latents outside the expected distribution. We adopt early-stopping for depth warping. As mentioned in Sec. 4.1, we disable the depth-based warping in the final 15 steps. Notably, blurred depth maps produced by simple heuristic filters inherit the same principle. Though not as good as out method, they still provide measurable improvements, as shown in the supplementary material. Together, these observations establish that dissolved depth maps offer a principled way to impose coarse geometric depth structure without the disadvantages of a high-frequency geometric depth prior.

**Limitations.** Our method exhibits strong robustness for videos at 256 and 512 resolutions, requiring minimal hyperparameter tuning. However, it may fail with high-resolution videos involving small objects under strong motion. In such cases, rapid fluctuations in the generated depth maps can cause instability in the latent space due to excessive and complex warping. To address this, the incorporated dissolved depth maps can help smooth artifacts and enhance overall performance. Nonetheless, this strategy demands careful tuning of dissolving levels (e.g., a stronger setting of 40 is recommended) and does not fully eliminate the issue. Residual inconsistencies in depth and semantic accuracy may persist, particularly in cases of extreme motion or occlusion, potentially leading to incorrect interpretations of depth and semantic information, as shown in Figure 11.

The observation from our user study and benchmark results is consistent with Tamir et al. (2024), which reveals that preferences in VR often differ from those observed on traditional screens. To be more specific, though *StereoCrafter* achieves a better epistolary consistency, *TrajectoryCrafter* delivers better stereo effects. In addition, *ProPainter* generates a great amount of artifacts on the occluded regions as shown in Fig. 12, which we anticipate would perform poorly in user studies. However, when viewed in stereo (e.g., with both eyes), many viewers did not notice the strong frame quality degradation of the

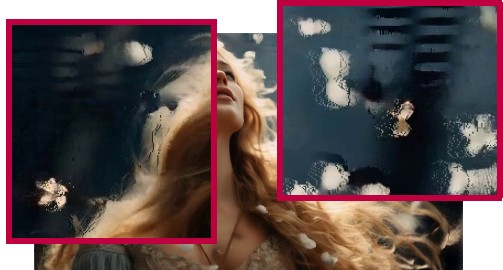

Figure 12: Artifacts generated by ProPainter.

right view. This discrepancy underscores the importance of developing evaluation metrics specifically tailored to immersive VR experiences, ensuring they accurately reflect perceptual feedback rather than relying on screen-based metrics.

## 5 CONCLUSION

In this work, we introduced *StereoCrafter-Zero*, a novel zero-shot stereo video generation approach. *StereoCrafter-Zero* incorporates a *noisy restart* strategy for stereo-aware latent initialization and an *iterative refinement* process to enhance latent consistency. We also proposed *dissolved depth maps* that retain only low-frequency structural depth, reducing high-frequency noise and improving the coherence and stability of the stereoscopic effects. Our comprehensive evaluations, including statistical analysis and user studies, demonstrate the effectiveness of our method in generating high-quality stereo videos with enhanced depth consistency and temporal smoothness. Future research will focus on developing an adaptive method for determining the optimal dissolving level and exploring the incorporation of user guidance for personalized control over the generated stereo videos.

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
