# OpenReview forum: "StereoCrafter-Zero: Zero-Shot Stereo Video Generation with Noisy Restart"
_ICLR.cc/2026/Conference — ICLR 2026 Conference Desk Rejected Submission_

### Official Review · Reviewer_4dv8 · 2025-10-31

**Soundness:** 3
**Presentation:** 3
**Contribution:** 3
**Rating:** 6
**Confidence:** 3

**Summary:**

This paper presents StereoCrafter-Zero, a zero-shot stereo video generation framework that synthesizes consistent stereo video pairs (left-right views) from a single image and text prompt, without requiring stereo training data. Previous work only involved generating stereo images, whereas this paper extends to video for the first time. The method builds upon video diffusion priors (e.g., DynamiCrafter) and introduces three core innovations:
1.	Noisy Restart — a latent initialization and controlled noise injection strategy to improve temporal and inter-view coherence;
2.	Iterative Refinement — repeated re-denoising of occluded regions to harmonize latent representations;
3.	Dissolved Depth Maps — low-frequency depth representations designed to enhance latent-space warping stability.

**Strengths:**

1.	The stereo video generation task remains underexplored yet highly relevant, addressing the growing demand for VR and immersive content creation. The paper provides a clear motivation for the necessity of joint spatial-temporal-stereo consistency, extending beyond previous single-view or 2D-to-3D conversion studies. The authors creatively identify a novel research problem and propose a training-free solution through StereoCrafter-Zero.
2.	Technical innovations are clear. The Noisy Restart strategy is conceptually elegant, reusing the diffusion process' stochasticity for structural stabilization. The Dissolved Depth Map idea is intuitive yet effective-reducing high-frequency depth noise aligns well with the latent-space nature of diffusion models. Iterative Refinement is a simple but practical scheme to correct occluded regions without excessive computational overhead.

**Weaknesses:**

1.	While the stereo video generation task is inherently complex, most evaluation metrics are adapted from standard video generation benchmarks, which lack stereo-specific evaluation criteria. Currently, only subjective user studies demonstrate the performance of StereoCrafter-Zero, which limits the objectivity of the validation. It is recommended to conduct quantitative comparisons with stereo diffusion models on established stereo conversion benchmarks to substantiate the approach further.
2.	The generated results exhibit certain artifacts. For example, in the teaser video, the left hand of Wukong shows spatial inconsistency, and the butterflies display temporal incoherence between left-right views. These issues highlight the challenges in maintaining stereo consistency for training-free frameworks.

**Questions:**

The authors compare against stereo conversion methods but not (or insufficiently) against established stereo generation or stereo-view synthesis benchmarks. Could the authors report results on one or more standard stereo datasets (or convert to them) to allow more objective quantitative comparison?

---

> ### Author Response · Authors · 2025-11-24
>
> * W1: We thank the reviewer for this suggestion. However, unlike the stereo conversion method that comes with a ground truth disparity or a ground truth right view to compare with, the generation method does not have such ground truth labels.
> Therefore, we use Met3r as metric to compare in the paper. As a generation method, we could not come up with a meaningful way to employ and compare our method for stereo conversion.
>
> * W2: The videos presented in the teaser are not adjacent frames since we would like to show stronger differences in the images. However, we acknowledge the challenges that the reviewer mentioned for a zero-shot method. Those challenges are precisely what we aim to address in our paper, and we have made important progress in solving them.

---

### Official Review · Reviewer_6Lsc · 2025-10-31

**Soundness:** 3
**Presentation:** 3
**Contribution:** 2
**Rating:** 6
**Confidence:** 3

**Summary:**

The paper proposes StereoCrafter-Zero, a zero-shot pipeline for stereo video generation from a single image and text prompt. The method keeps left/right views temporally coherent by operating in the latent space of a video diffusion model, with key techniques combining:
1. Noisy Restart to initialize stereo-aware latents
2. Iterative Refinement to denoise the occluded regions
3. Dissolved (low-frequency) depth maps to stabilize latent warping without relying on precise disparities.

On the proposed benchmarks, the proposed method attains the best quantitative performance and strong user-study scores for frame quality and temporal coherence.

**Strengths:**

1. The paper is with clear problem framing and novelty. It's very intuitive and beneficial to tie the generation and stereo conversion within diffusion latents rather than pixels. To enable the combination of low-frequency information and high-frequency details.

2. Noisy Restart and Iterative Refinement components are simple, well-motivated, and ablated extensively in the experiment section.

3. And the authors have conducted extensive experiments to show the superior performance of the proposed method, and the validness of each proposed components.

**Weaknesses:**

1. The proposed pipeline involves depth estimation, repeated warping, and multiple diffusion passes. An end-to-end latency and compute report and the comparisons against baselines is needed, and current not systematically reported in the paper.

2. Intuitively, the proposed dissolved depth operates on the latent space of the diffusion model, which is with naturally downsampled space. This seems especially bad for high-resolution details and small objects. It'd be good to at least provide a qualitative analysis upon this perspective.

3. Authors note mismatches between screen metrics and VR preference. Since currently VR-related metrics are still not well established, it'd be good to provide some qualitative examples to support this claim, and make the readers be more aware of the desired direction for future research.

**Questions:**

Mostly from the above weakness part, some additional questions:

1. What are the wall-clock times and memory usage for a typical video, including depth + diffusion + refinements? How do Noisy Restart window K/L and refinement rounds N trade off quality vs. runtime and memory cost?

2.How does performance change when swapping the current video model with other backbones? Any failure trends across certain content types, e.g. human, animals?

---

> ### Author Response · Authors · 2025-11-24
>
> * W1/Q1: We thank the reviewer for the suggestion. As detailed in Tables 4 and 5 of the main paper, we already reported the impact of the Noisy Restart window K/L and refinement rounds N on the quality. The results indicate an optimal threshold for these settings, beyond which additional increases yield no further quality improvements. Note that the parameter of "total steps" can be seen as a proxy for the runtime for the number of diffusion steps.
> To expand on this, we have compiled the following new table for wall-clock times and memory usage across resolutions. Note that TrajectoryCrafter and StereoCrafter are limited to specific resolutions, and our implementation is not fully optimized yet. Future optimizations, like latent reuse, could reduce times to about 2/3, making it comparable to TrajectoryCrafter. We evaluate on a single NVIDIA A100 GPU.
>
>
>     |Res        | TrajectoryCrafter      | StereoCrafter         | Ours                  |
>     |-----------|------------------------|-----------------------|-----------------------|
>     | 1024      | 193.87 (s), 27.61 (GB) | -                     | 344.19(s), 22.92 (GB) |
>     | 512       | -                      |  49.95(s), 6.48 (GB)  | 107.80(s), 17.91 (GB) |
>     | 256       | -                      |   -                   |  54.32(s), 16.96 (GB) |
>
>
> To be more specific, on all those resolutions, the running time for depth maps is around 2\% of the total runtime. The running time for the warping operation can be ignored by using the proposed efficient warping algorithm that is reported in our supplementary material.
>
> * W2: We thank the reviewer for raising this point. Our dissolved depth indeed operates on the naturally downsampled latent space. However, there are more channels per pixel (token), so that this downsampled space is still capable of representing high-resolution details.
> In case the reviewer is concerned about warping depth maps only at early diffusion stages, we provide an answer to this question in our comments to W2 of Reviewer 2.
>
> We demonstrated scenarios with small objects such as raindrops and flying flowers, in which we do not observe strong misalignment or failures. As reported in our paper, we observe stronger artifacts for videos with large motion (see Figure. 11), instead of high-resolution details and small objects.
>
>
> * W3: We value the reviewer's point on the mismatch between screen metrics and VR preferences, given the lack of established VR metrics. For instance, from our experiments, ProPainter generates a great amount of artifacts in the occluded regions, which we anticipate would perform poorly in user studies. However, when viewed in stereo (e.g., with both eyes), many viewers did not notice the strong frame quality degradation of the right view. It seems our human vision system can fix the bad regions automatically. Thus, we assume the depth cues' precision takes over the inpainting quality in real stereo video watching experiences. We included more explanation and a visual example in our limitations section.
>
> * Q2: In the supplement (section C.3), we illustrate some results when using the latent video diffusion models (LVDMs) as a backbone, which reveals a bad choice in comparison to DynamiCrafter in terms of the noise pattern handling. From our tests, we found that the generation of human body and facial expressions can be problematic, especially with large motions. This is an inherited issue from DynamiCrafter, as discussed in the GitHub issue on "How to fix the human body or those incorrect generation?" (https://github.com/Doubiiu/DynamiCrafter/issues/127).
> We assume that newer video generation models will improve upon this issue.

---

### Official Review · Reviewer_T9o9 · 2025-11-01

**Soundness:** 2
**Presentation:** 2
**Contribution:** 2
**Rating:** 2
**Confidence:** 4

**Summary:**

The paper presents StereoCrafter-Zero, a method for zero-shot stereo video generation from a single image and text prompt. It introduces Noisy Restart and Iterative Refinement to improve stereo and temporal coherence, along with Dissolved Depth Maps to reduce noise.

**Strengths:**

The paper introduces a novel framework for zero-shot stereo video generation, combining techniques like Noisy Restart and Iterative Refinement to enhance stereo consistency and temporal coherence without the need for paired training data.

**Weaknesses:**

1. The video results presented do not include significant camera movements, and the camera remains almost static. This setup does not effectively demonstrate the performance of the proposed method. In real-world scenarios, large camera movements are common, and it is questionable how well the proposed method would perform under such conditions.

2. Given the artifacts in the results from DepthCrafter, it is unclear why the authors chose it as the prior model instead of VideoDepthAnything, which demonstrates much better performance. Additionally, the claim in Line 268 that high-precision depth leads to artifacts, as well as the comparison in Table 3 with different depth estimation methods, is confusing and weird. If lower-quality depth maps can lead to better results, this should be clarified with more explanation or supporting evidence instead of suspection.

3. The paper claims that latent space warping benefits more from coarse geometry than fine depth, and unlike image-domain warping, latent space warping focuses on coarse geometry and semantic consistency rather than relying on high-precision depth maps. However, how does the method ensure fine-grained alignment of details? The authors demonstrate simple examples, but how does it perform with more complex fine details? Is the use of lower-quality depth maps a trade-off for better high-precision details, or is there another reason for this choice?

**Questions:**

See the weakness part.

---

> ### Author Response · Authors · 2025-11-24
>
> * W1: We acknowledge the reviewer's observation regarding the absence of significant camera movements in our video results.
> To clarify, our approach is actually capable of handling significant camera movements, as we have explored in additional experiments. We acknowledge that the current demonstrations do not adequately showcase this, and we have prepared examples featuring larger camera motions. More videos with larger camera and object motions are now included in the supplementary material.
>
> * W2: We thank the reviewer for raising concerns about our selection of DepthCrafter and the associated claims on depth maps.
> For the first part of the question: As shown in Table 8 of the supplementary materials, we have included experiments with VideoDepthAnything, DepthAnything, and Depth Pro for comparison. Our method is compatible with any depth method. However, the depth maps are processed by a depth dissolving technique. This technique is implemented as the reverse diffusion process in the diffusion latent space. This reverse diffusion process comes from the pre-trained Depthcrafter architecture. We conjecture that this makes the distribution of depth latents from DepthCrafter more compatible, resulting in significant performance improvements as demonstrated in Table 8 (4.96 vs. 5.91 from VideoDepthAnything).
> For the second part of the question:
> To clarify, our use of dissolved depth maps does not imply that lower-quality maps are superior. We argue that warping depth maps at earlier diffusion stages only is better than warping depth maps at all diffusion stages. This is experimentally verified.
> It is intuitive that a warping operation alters the distribution of RGB video latents to a distribution that is slightly out of the domain for the diffusion model.
> Therefore, there is a trade-off between warping high-frequency details at later diffusion stages and moving latents outside the expected distribution.
> To clarify, we added more discussion in Section 4.3 of the paper.
>
>
> * W3: Our answer to this question is similar to the previous answer W2. Warping has a benefit in earlier diffusion stages. If warping is used at later diffusion stages, the latents will be slightly out of the domain, and the diffusion model does not have sufficient ability to correct the latents and ensure high-quality visual details.
> There is a trade-off between ensuring high visual quality and enforcing the consistency of high-frequency depth details. If one attempts to guarantee the consistency of high-frequency depth-details, the problem becomes over-constraint and ultimately, both visual quality and depth consistency end up being much worse.
> We demonstrated scenarios with complex fine details such as fire flames, raindrops, and flying flowers, in which we do not observe strong misalignment or failures. As reported in our paper, we observe stronger artifacts for videos with large motion (see Figure. 11), instead of complex fine details.

---

### Official Review · Reviewer_SeqF · 2025-11-01

**Soundness:** 3
**Presentation:** 3
**Contribution:** 2
**Rating:** 6
**Confidence:** 5

**Summary:**

The paper proposes a training-free stereo video generator from a single image + prompt by coupling a pre-trained video diffusion prior with three sampling-time mechanisms: Noisy Restart (re-inject noise at selected steps), Iterative Refinement (masked denoising on occluded regions), and Dissolved Depth Maps (low-frequency depth to guide latent warps). The method optimizes stereo consistency in latent space rather than pixel space and reports gains on MEt3R (epipolar consistency) and a small VR user study.

**Strengths:**

1. The stereo content generation is an important but under-explored application domain. And the angle of generating stereo content generation in a training-free manner is neat.
2. The latent-space stereo coupling (warp + masked refinement) is clean and avoids heavy training or pixel-space inpainting, the dissolved-depth idea is a reasonable inductive bias at sampling time.
3. The evaluation is comprehensive with Stereoscopy-aware metrics, which is important for stereo content generation. The performance is state-of-the-art.

**Weaknesses:**

1. The core technical contribution "Noisy Restart" mechanism is essentially a sampling-time stochasticity strategy. Conceptually, it closely parallels the Noise Re-Injection approach introduced in Time Reversal Fusion (Explorative Inbetweening of Time and Space, Feng et al. ECCV 2024), where noise is periodically reintroduced during denoising to stabilize bidirectional generation trajectories. While this work applies the idea in a stereo-latent context rather than bounded in-betweening, the underlying principle—injecting mid-trajectory noise to prevent degeneracy and enhance temporal smoothness—is largely shared. I think the authors should explicitly acknowledge this lineage, clarify what aspects are unique to the stereo setting (e.g., the interaction with disparity-guided warping and dissolved-depth refinement).
2. The stereo effect is largely affected by the latent-space depth prediction and warping, I wonder how would the method work given common failure modes of monodepth model, like volumetric / transparent cases (flame, raindrops, glass)
3. I think the proposed method should be compared with more stereo video generation baselines like monocular video to stereo generation (such as SpatialDreamer Lv et al. CVPR 2025).

**Questions:**

I would like to know the authors thoughts regarding my concerns in the weakness section.

---

> ### Author Response · Authors · 2025-11-24
>
> * W1: We thank the reviewer for proposing this relevant reference.
> We agree that our Noisy Restart mechanism shares the high-level idea of reintroducing noise to influence the generation at certain diffusion steps, and iterating this process several times before advancing to the subsequent denoising step.
> The key difference is that the Time Reverse Fusion (TRF) paper injects noise into the entire fused latent, with a focus on global harmony. While our method is specifically tailored to stereo-aware generation. It injects noise only into the “filling” region of the target view latent (occluded / disoccluded areas resulting from disparity-guided warping). We included the proposed reference in our discussion section of "Noise-Injection For Latent Refinement".
>
> * W2: We thank the reviewer for raising this insightful question regarding the robustness of our method to common failure modes of monodepth models. Based on our experiments, the model has demonstrated strong performance on cases involving fire flames and raindrops, which we attribute to the advantages of performing warping in the latent space. This latent-space approach provides inherent error tolerance by mitigating the propagation of depth prediction inaccuracies, allowing for more stable stereo generation.
>
>
>     In addition, we evaluate the depth failure cases within the examples from the MonoTrap dataset (https://arxiv.org/abs/2412.04472). By utilizing the input images from this dataset, our stereo generation model is able to reconstruct physically plausible videos despite the misleading depth cues from the conditioning image. We included relevant videos in our supplementary material.
>
>
> * W3: We thank the reviewer for their suggestion. However, we have not yet identified any open-sourced methods for stereo generation that are readily available for direct evaluation. Since the code of SpatialDreamer (Lv et al. CVPR 2025) is not currently open-sourced, we are unable to include it in our comparisons at this time.

---

### Author Response · Authors · 2025-11-25

Dear Reviewers and Area Chair,

We would like to inform you that we have updated our submission to incorporate all recent revisions. In particular, we have added additional video examples to the supplementary material to better illustrate the qualitative behavior of our method. To assist your review, we have also included a PDF-diff version of the main paper in the supplementary material, highlighting all changes made since the previous submission.

We sincerely appreciate your time and constructive feedback, and we hope these updates help clarify and strengthen our work.

Thank you again for your consideration.

Best regards,
The Authors

---

### Author Response · Authors · 2025-11-30

Dear Reviewers and Area Chair,

We provide a summary of our paper and rebuttal below.

Three out of four reviewers (SeqF (R1), 6Lsc (R3), 4dv8 (R4)) rated the paper as "Marginally Above Acceptance" (6), while one reviewer (T9o9 (R2)) recommended rejection (2).

There is a strong consensus on the novelty and significance of the work:

- **Underexplored Problem**: Reviewers agree that extending zero-shot generation from stereo images to stereo videos is a significant and timely contribution (SeqF (R1), 4dv8 (R4)).

- **Methodological Elegance**: The latent-space warping and "Noisy Restart" mechanisms are praised as intuitive and clean solutions that avoid the heavy computational cost of training or pixel-space inpainting (SeqF (R1), 6Lsc (R3)).

- **Evaluation**: The use of stereoscopy-aware metrics (e.g., MEt3R) was noted as a strength (SeqF (R1)).

The primary contention comes from Reviewer T9o9 (R2), who raised concerns about camera motion magnitude and the counter-intuitive use of "Dissolved Depth" (lower frequency depth maps). We have provided a rebuttal that addresses these points, alongside the concerns of the positive reviewers.

1. **Addressing Motion & Depth** (Re: Reviewer T9o9 (R2)):
    - Critique: T9o9 (R2) felt the demos lacked camera movement and questioned why high-precision depth wasn't used.
    - Resolution:
        - We added supplementary videos demonstrating larger camera motions, directly refuting the limitation claimed by the reviewer.
        - We clarified that high-frequency depth warping at later diffusion stages pushes latents out of distribution, degrading visual quality. Meanwhile, during its initial stages, the diffusion process only focuses on retrieving the main structures of the scene. Thus, "Dissolved depth" is a deliberate design choice to balance geometric guidance with the diffusion prior's domain.

2. **Technical Novelty vs. Prior Work** (Re: Reviewer SeqF (R1)):
    - Critique: SeqF (R1) noted similarities between "Noisy Restart" and "Time Reversal Fusion" (TRF).
    - Resolution: We acknowledged the lineage but clarified the distinction: TRF targets global harmony for in-betweening, whereas Noisy Restart is tailored for stereo-specific "filling" regions (occlusions). This is now discussed in the revised paper.

3. **Computational Cost** (Re: Reviewer 6Lsc (R3)):
    - Critique: Lack of wall-clock time analysis.
    - Resolution: We provided a detailed runtime/memory table. We clarified that the current implementation prioritizes generation quality over inference speed to establish a strong baseline for this novel task. We proposed a clear optimization path by reusing latents rather than recomputing them, which would bring inference time in line with standard baselines (TrajectoryCrafter) without sacrificing the demonstrated quality.

**Conclusion**: We have effectively addressed all the reviewers' concerns, including those raised by reviewer T9o9 (R2) regarding motion magnitude and depth strategy, through additional experiments and theoretical clarification. Given the high difficulty of zero-shot stereo video generation and the consensus on the method's novelty and soundness among the majority of reviewers, we believe this paper merits acceptance. Moreover, the approach represents a solid baseline for future training-free stereo generation research.

---

### Note · Program_Chairs · 2026-01-17
**Submission Desk Rejected by Program Chairs**

The following references in this submission do not refer to real documents and/or have major errors in bibliographic information:

 Jonathan T Barron and Jovan Popović. Structure-from-motion with oriented points. In IEEE Transactions on Pattern Analysis and Machine Intelligence, 2015.